# A Patient with Corticobasal Syndrome and Progressive Non-Fluent Aphasia (CBS-PNFA), with Variants in *ATP7B*, *SETX*, *SORL1*, and *FOXP1* Genes

**DOI:** 10.3390/genes13122361

**Published:** 2022-12-14

**Authors:** Katarzyna Gaweda-Walerych, Emilia J. Sitek, Małgorzata Borczyk, Ewa Narożańska, Bogna Brockhuis, Michał Korostyński, Michał Schinwelski, Mariusz Siemiński, Jarosław Sławek, Cezary Zekanowski

**Affiliations:** 1Department of Neurogenetics and Functional Genomics, Mossakowski Medical Research Institute, Polish Academy of Sciences, 02-106 Warsaw, Poland; 2Laboratory of Clinical Neuropsychology, Neurolinguistics and Neuropsychotherapy, Division of Neurological and Psychiatric Nursing, Faculty of Health Sciences, Medical University of Gdansk, 80-211 Gdansk, Poland; 3Neurology Department, St. Adalbert Hospital, Copernicus PL, 80-462 Gdansk, Poland; 4Laboratory of Pharmacogenomics, Department of Molecular Pharmacology, Maj Institute of Pharmacology Polish Academy of Sciences, 31-343 Krakow, Poland; 5Department of Nuclear Medicine, Faculty of Health Sciences, Medical University of Gdansk, 80-214 Gdansk, Poland; 6Neurocentrum Miwomed, 80-207 Gdansk, Poland; 7Division of Neurological and Psychiatric Nursing, Faculty of Health Sciences, Medical University of Gdansk, 80-211 Gdansk, Poland; 8Department of Emergency Medicine, Faculty of Health Sciences, Medical University of Gdansk, 80-210 Gdansk, Poland

**Keywords:** corticobasal syndrome, progressive non-fluent aphasia (CBS-PNFA, CBS-NAV), whole exome sequencing (WES), *ATP7B*, *SETX*, *SORL1*, *FOXP1*, splicing analysis, mitochondrial network analysis

## Abstract

Our aim was to analyze the phenotypic-genetic correlations in a patient diagnosed with early onset corticobasal syndrome with progressive non-fluent aphasia (CBS-PNFA), characterized by predominant apraxia of speech, accompanied by prominent right-sided upper-limb limb-kinetic apraxia, alien limb phenomenon, synkinesis, myoclonus, mild cortical sensory loss, and right-sided hemispatial neglect. Whole-exome sequencing (WES) identified rare single heterozygous variants in *ATP7B* (c.3207C>A), *SORL1* (c.352G>A), *SETX* (c.2385_2387delAAA), and *FOXP1* (c.1762G>A) genes. The functional analysis revealed that the deletion in the *SETX* gene changed the splicing pattern, which was accompanied by lower *SETX* mRNA levels in the patient’s fibroblasts, suggesting loss-of-function as the underlying mechanism. In addition, the patient’s fibroblasts demonstrated altered mitochondrial architecture with decreased connectivity, compared to the control individuals. This is the first association of the CBS-PNFA phenotype with the most common *ATP7B* pathogenic variant p.H1069Q, previously linked to Wilson’s disease, and early onset Parkinson’s disease. This study expands the complex clinical spectrum related to variants in well-known disease genes, such as *ATP7B*, *SORL1*, *SETX*, and *FOXP1*, corroborating the hypothesis of oligogenic inheritance. To date, the *FOXP1* gene has been linked exclusively to neurodevelopmental speech disorders, while our study highlights its possible relevance for adult-onset progressive apraxia of speech, which guarantees further study.

## 1. Introduction

Corticobasal syndrome (CBS) and progressive non-fluent aphasia (PNFA) are predominantly sporadic syndromes [1,2]. Only c.a. one-third of CBS cases are caused by a genetic mutation in *MAPT*, *PGRN*, *LRKK2*, *PSEN1*, or *C9orf72* genes [3,4,5,6], reviewed in [2,7,8], with a few predisposing factors, including *MAPT* H1 haplotype [9,10].

PNFA is usually associated with frontotemporal lobar degeneration (FTLD), where monogenic mutations account for about one-third of cases [11]. The PNFA phenotype has shown the strongest association with *PGRN* mutations [12,13,14]. Patients diagnosed clinically with distinct syndromes from the FTLD spectrum typically show symptoms of other syndromes later in the disease course. PNFA patients frequently develop CBS while individuals diagnosed initially with CBS are likely to present with non-fluent aphasia later in the disease course [15,16]. Apraxia of speech and non-fluent aphasia have (relatively recently) been acknowledged as parts of CBS [17]. CBS is characterized by strongly asymmetric presentation, comprising of at least 2 among the following motor features: (a) limb rigidity or akinesia, (b) limb dystonia, (c) limb myoclonus. Additionally, at least 2 other features should be present: (a) orobuccal or limb apraxia, (b) cortical sensory deficit, (c) alien limb phenomena [18,19]. Current criteria of corticobasal degeneration (CBD) specify the variant with non-fluent/agrammatic progressive aphasia (CBD-NAV) as one of the possible disease phenotypes [18]. This variant is associated with predominant dysfunction of the left-hemisphere and predominantly right-sided motor symptoms [18].

Of note, there have been very few genetic reports regarding the overlapping CBS-PNFA phenotype. To date, one case of symmetrical CBS with non-fluent aphasia caused by a *PGRN* mutation [20], two cases with CBS associated with speech apraxia caused by a *MAPT* mutation [21,22] and two cases sharing CBS and PPA features with a *LRRK2* mutation [6] were described. However, genetic screening for *PGRN* and *MAPT* mutations was negative in a group of nine patients with PNFA associated with CBD pathology [23], suggesting that most of these overlapping phenotypes may not be monogenic, which highlights the need to look for oligogenic etiology. The implementation of whole exome/whole genome sequencing (WES/WGS) technology highlighted a complex genomic architecture of disease phenotypes, inciting researchers to revisit the classical concepts of genetic causality and monogenic disorders [24]. The evidence that oligogenic inheritance may account for a substantial fraction of atypical and unresolved cases of neurodegenerative diseases is mounting [25,26,27]. Indeed, it is estimated that c.a. 50% of sporadic patients from the FTLD spectrum carry at least one rare missense variant in neurodegenerative disease-associated genes [28]. Although there are detailed clinical and research criteria for FTLD syndromes, the clinical phenotype frequently evolves in the disease course often comprising cross-disease symptoms of variable intensity. Thus, from the point of view of genetic testing, next-generation sequencing is more adequate for the detection of the genetic variants contributing to overlapping phenotypes.

Our aim was to analyze the phenotypic-genetic correlations in a patient diagnosed with early onset CBS-NAV with predominant apraxia of speech. As polygenic etiology may be of particular relevance in overlapping dementia syndromes with early onset [29], we conducted whole exome sequencing (WES).

## 2. Materials and Methods

### 2.1. Whole Exome Sequencing (WES) Data Preprocessing

WES was performed commercially by CeGaTGmbH|Paul-Ehrlich-Str. 23|D-72076 Tübingen|Germany. The following library preparation kit was used: Twist Human Core + RefSeq + Mitochondrial Panel (Twist Bioscience); 92.37% percent of sequenced bases had a predicted quality score of 30 or more (Q30 value). Sequencing parameters were NovaSeq 6000; 2 × 100 bp. Demultiplexing of the sequencing reads was performed with Illumina bcl2fastq (2.20). Statistics of mapped reads are provided in Table 1.

Raw read files were then processed with Intelliseq Flow Annotation Pipeline (https://intelliseq.com/ accessed on 25 August 2022) built in Cromwell (https://cromwell.readthedocs.io/en/stable/ accessed on 25 August 2022). Within the pipeline, fastq file quality was assessed with FastQC. Fastq files were then aligned to the Broad Institute Hg38 Human Reference Genome with GATK 4.0.3. Duplicate reads were removed with Picard and base quality Phred scores were recalibrated using GATK’s covariance recalibration. Variants were called using GATK best practices. Identified variants were assessed using Intelliseq Flow annotation pipeline that implements the American College of Medical Genetics and Genomics (ACMG) recommendations.

In short variant annotation and annotation-based filtering, 3182 genes were included in the screening based on Human Phenotype Ontology (HPO) terms (Appendix A). Databases used for annotation included gnomAD v2.1 and v3 (frequencies, coverage, constraint), 1000Genomes (frequencies), MITOMAP (frequencies, contributed diseases), ClinVar (contributed diseases, pathogenicity), Human Phenotype Ontology (HPO) terms (inheritance mode, contributed phenotypes and diseases), UCSC (repeats, PHAST conservation scores), SIFT4G (constraint), SnpEff (predicted impact on gene product), dbSNP (rsID), Ensembl (gene and transcript information), and COSMIC (somatic mutations data). Common and low-impact variants (with a max frequency threshold of 0.05 and minimal SnpEff predicted impact on gene product set as MODERATE) were then removed. Variants with the following impacts on protein and mRNA were retained: missense, nonsense, frameshift, and splice site variants. At this step, 711 variants remained (Appendix A). Annotated variants were then classified according to the ACMG criteria and prioritized, yielding 164 variants (Appendix A). Selected variants were manually evaluated for quality in IGV and confirmed in Sanger sequencing (Appendix A
*ATP7B*, Appendix A
*SETX*).

The *C9orf72* expansion analysis was performed using previously described protocols [30] using Genetic Analyzer 3130 and SeqScape v2.5 software (Applied Biosystems, Foster City, CA, USA).

### 2.2. Fibroblast Cultures and Inhibition of Nonsense-Mediated Decay (NMD)

Primary skin fibroblasts (obtained from the patient and age-matched, unrelated, neurologically healthy subjects) were collected and cultured as previously described [31]. For inhibition of nonsense-mediated mRNA decay (NMD), primary fibroblast cultures were treated with an NMD inhibitor, cycloheximide (500 µg/mL), or DMSO for 10 h (Sigma Aldrich, Saint Louis, MO, USA), as previously described [31].

### 2.3. Reverse Transcription, mRNA Isoform Analysis, and Real-Time PCR (RT-PCR)

RNA was extracted according to standard protocol with QIAzol Lysis Reagent (Qiagen, Manchester, UK) and reverse transcribed with cDNA synthesis kit, NG dART RT (EURx Molecular Biology Products, Gdańsk, Poland; cat. no. E0801), using random hexamers primers [32]. For the PCR, AllTaq Master Mix Kit was used (Qiagen, Manchester, UK). The real-time PCR analysis was conducted with RT HS-PCR Mix SYBR (A&A BIOTECHNOLOGY, Gdańsk, Poland) using a StepOne Plus system (Applied Biosystems, Foster City, CA, USA). Changes in gene expressions were determined with the DeltaCt method using *GAPDH* levels for normalization. Similar results were obtained with *PPIB* as a housekeeping gene. Primers are listed in Appendix A. PCR products were run on 2% agarose gel, selected bands were excised from the gel and extracted with QIAquick Gel Extraction Kit (Qiagen, Manchester, UK). Sanger sequencing was performed by (https://oligo.ibb.waw.pl accessed on 2 September 2022).

### 2.4. Mitochondrial Network Staining and Analysis

Patient fibroblasts (P) and control fibroblasts (C1 and C2) were grown on glass coverslips in standard conditions for 24 h [31]. For mitochondrial staining, cells were washed twice with Phosphate Buffered Saline, fixed with 4% paraformaldehyde (PFA) for 10 min, blocked with Normal Goat Serum (Thermo Fisher Scientific Inc., Waltham, MA, USA; cat no. 50197Z) for one hour, incubated overnight with rabbit anti-Hsp60 (*HSPD1*, Heat Shock Protein Family D (Hsp60) Member 1) antibody (1:500; Abcam, Cat No. ab53109) to visualize mitochondria [33], followed by one hour of incubation with secondary anti-rabbit antibody Alexa Fluor−555 (Thermo Fisher Scientific Inc., USA; Cat No. A-31570) and 10-min incubation with DAPI to visualize cell nuclei. Images were obtained by Zeiss LSM 510 confocal microscopy under 63x objective. Mitochondrial morphology parameters, including Form Factor (reflecting mitochondrial length) and Aspect Ratio (reflecting both the length and the extent of mitochondrial branching), were assessed with ImageJ as described in [34,35,36].

### 2.5. Statistical Analysis

For each experiment, the relative values obtained from different biological replicates. (n = 3) were used to calculate means, medians, standard deviations (SD), and statistical significance. The Kruskal–Wallis test and two-tailed unpaired *t*-test with Welch’s correction were used (GraphPad Prism 6.0). *p* < 0.05 was considered significant.

## 3. Results

### 3.1. Case Description

The patient was followed clinically by our team from the age of 66 until her death at the age of 72. Her medical history included type 2 diabetes, hypertension, and gout. At the age of 63, she experienced minor problems when parking her car (distance judgment) (Appendix A. Case timeline). Later, at the age of 64, she noticed speech problems (possibly apraxia of speech) and fell from a chair on wheels. Soon afterward she left her job due to speech problems. PNFA was diagnosed at the age of 66. Her speech was very effortful and non-fluent. The pattern of speech impairment indicated the predominance of speech apraxia, but mild agrammatism was also present and more evident in writing. Spontaneous falls became more frequent. She dropped objects and presented with a strong left-sided preference when navigating the environment, suggestive of right-sided unilateral neglect. She was unable to tie shoelaces, zip her jacket, and struggled to use cutlery, which corresponded to apraxia. Her poor decision-making was evidenced by her succumbing to advertising. She had preserved insight into communication problems. Apathy, mild impulsivity, and irritability were also present. Her handwriting with her right hand was extremely effortful so she switched to writing on a keyboard using her left hand. Of note, she used to be left-handed and had been forced to write with her right hand. Moreover, she used to play the accordion and her left finger dexterity could have been above average. The patient was diagnosed with possible CBD with a non-fluent/agrammatic aphasia variant (NAV) phenotype [18].

At the age of 67, she became depressed. When she was 69, her gait deteriorated further and she demonstrated apraxia in her right lower limb. At the age of 70, she walked only with assistance, her speech was limited mainly to yes/no answers, she could not initiate movement with her right upper limb, and swallowing problems were evident.

When she was 72, she was unable to maintain a sitting position and dysphagia was severe. She died a few months later at a nursing home. Proband’s father presented symptoms similar to those observed in our patient (tremors, loss of speech), at the age of 68, four years before his death.

#### 3.1.1. Clinical Assessment

The patient underwent her first neurological examination at the age of 67. Non-fluent aphasia was the predominant syndrome with the right upper limb apraxia, involuntary mirror right limb movement, parkinsonian tremor, rigidity, and dystonia. The patient had postural instability with a tendency to fall forward. The symptoms were not levodopa responsive. With disease progression, predominantly right-sided spastic dystonia, facial dystonia, and spastic dysarthria appeared. The patient was assessed neuropsychologically four times (see: Appendix A, see also Figure 1C,D and Appendix A). At the age of 67, the patient presented with apraxia of speech, non-fluent aphasia, aprosody, right-sided unilateral neglect, ideomotor apraxia, mirror movements, synkinesis (e.g., when the patient tried to write with her right hand, lip and right leg movements were observed), alien limb phenomenon, and mild cortical sensory loss, were accompanied by executive deficits. At this time, motor hemispatial neglect was predominant. During alternate bilateral hand movements, there was reduced amplitude, and then extinction in the right hand. The severity of deficits corresponded to mild cognitive impairment as the patient’s disability was mainly due to motor impairment and aphasia. A year later, apraxia and alien limb phenomenon extended to the right lower limb and hemispatial neglect was more pronounced in visual search tasks. At the age of 68, there was a marked deterioration of language; agrammatism was present both in speech and writing. The patient was unable to perform a pegboard task with the right hand. Moreover, her tactile object recognition and finger recognition deteriorated, and cortical sensory loss was more prominent, especially when the right hand was tested. Her cognitive status progressed to mild dementia. At the last examination, when she was 69, she could hardly repeat two-syllable words and perform a simple pegboard task with her left hand due to marked hypokinesia and difficulty in moving pegs. Her visual episodic memory was still quite well preserved at this time.

#### 3.1.2. Laboratory Testing

Repeated standard laboratory blood tests, including alanine and aspartate transferase, and an ultrasonography of the abdomen with the liver assessment were normal.

The evoked autonomic sympathetic skin potentials were within normal range, but the heart rate variability was high. Electroencephalography was normal. Electroneurography revealed no clinically significant changes.

#### 3.1.3. Neuroimaging

Magnetic resonance imaging (MRI) of the head performed at the age of 67 demonstrated mild cortical atrophy of both frontal lobes (Figure 1A). The subtentorial atrophy did not exceed the normal range. Single photon emission computed tomography (SPECT) revealed bilateral frontal and left parietal hypoperfusion (Figure 1B).

#### 3.1.4. Genetic Assessment

Whole exome sequencing (WES) of the patient’s DNA revealed 164 rare variants classified according to the American College of Medical Genetics and Genomics (ACMG) criteria for pathogenicity (Appendix A) [37].

We focused on the rare variants with the highest putative impact on the disease phenotype (Table 2). The presence of *C9orf72* expansion was excluded [30]. Among common risk variants, the patient had the *APOE* ε2, ε2 genotype.

According to the Human Splicing Finder prediction, the *SETX* variant c.2385_2387delAAA in exon 10 activates a cryptic donor site, which may lead to splicing alteration (Appendix A). cDNA from the patient’s fibroblasts (Figure 2A, lane 6, P) had an altered pattern of exon 10 splicing with an additional aberrant isoform (band *A), visible only upon inhibition of nonsense-mediated decay mechanism with cycloheximide (CHX) treatment. Sequencing of the band *A, excised from agarose gel (Figure 2A, line 6, band *A) confirmed that band A corresponds to splicing isoform with 358 bp of exon 10 skipped (Figure 2C, Appendix A). The skipping of the 358 bp fragment led to the frameshift and introduced a premature termination codon (PTC). The skipped fragment had canonical splicing consensus sequences GT-AG (donor and acceptor, respectively). However, the new alternative donor site (used during the splicing event) was located downstream of the sites predicted by HSF, which highlighted the importance of performing a functional analysis since in silico predictions may not be precise.

As a positive control, to confirm correct NMD inhibition, we used mRNA transcribed from the *DSCR1* gene (the Down syndrome critical region 1), a known NMD target [39,40]. As expected, *DSCR1* levels increased upon cycloheximide treatment, showing robust NMD inhibition (Figure 2B).

Sequencing of the excised band *B (Figure 2A, lane 6) revealed that it is an artifact. The sequencing of the main band (Figure 2A, line 6) confirmed 100% identity with the expected PCR product—645 bp fragment of *SETX* exon 10 with c.2385_2387delAAA variant (Appendix A). In parallel, a real-time analysis showed lower levels of *SETX* mRNA (Figure 2D, see also Appendix A).

#### 3.1.5. Mitochondrial Network Assessment

Since *SORL1, ATP7B, SETX*, and *FOXP1* genes have been previously linked to mitochondrial dysfunction [41,42,43,44], we performed mitochondrial staining in the patient’s fibroblasts with Hsp60, a mitochondrial chaperonin essential for protein folding and assembly in the mitochondrial matrix [33]. We observed a particular network shape in the patient with significantly altered mitochondrial parameters compared to neurologically healthy controls (Figure 3A–C). The lower aspect ratio in the patient indicated decreased mitochondrial connectivity (Figure 3B), while a lower form factor pointed to shorter (and less branched) mitochondria, compared to CTRLs (Figure 3C).

## 4. Discussion

The variability of CBS-NAV clinical phenotype suggests the important contribution of oligogenic background in disease development [24]. Detection of rare, putatively functional genetic variants linked to similar clinical phenotypes would corroborate this hypothesis.

We discuss below how the rare genetic variants detected in the patient (Table 2) could have contributed to the observed phenotype in light of the up-to-date literature and our clinical, and genetic analyses.

### 4.1. Complex Genetic Landscape

The *ATP7B* gene encodes a Cu-transporting P-type ATPase protein responsible for incorporating copper into apo-ceruloplasmin transporting copper in the serum. Homozygous or compound heterozygous mutations in the *ATP7B* gene caused Wilson disease (WD), characterized by excessive copper accumulation in various organs, such as the brain, liver, and cornea, leading to a wide spectrum of symptoms from hepatic to neuro-psychiatric [45,46,47,48]. The biochemical hallmarks of WD are decreased ceruloplasmin and elevated copper levels in blood serum.

In most WD patients, mutations are clustered in exon 14 of the *ATP7B* gene [49,50] that encodes part of the ATPase nucleotide-binding domain, responsible for ATP binding and enzyme activity [51]. The most common *ATP7B* pathogenic variant H1069Q in exon 14 accounts for 72% of WD cases in the Polish population [52,53]. Very rarely, carriers of single heterozygous *ATP7B* mutations (Hzgs) demonstrate clinical symptoms, and biochemical and/or brain abnormalities [54,55] (Appendix A). However, recently, more disease phenotypes have been described in *ATP7B* Hzgs, such as early or late-onset Parkinson’s disease (EOPD or LOPD) (summarized in Appendix A). Given the mounting evidence of the possible involvement of single heterozygous *ATP7B* mutations in phenotypes with parkinsonism (Appendix A), it can be speculated that p.H1069Q mutation in our CBS-NAV patient has led to the subclinical dysregulation of copper metabolism. Motor symptoms that were present in our patient included dystonia, rigidity, and rest tremor, and they initially impaired the function of the right limbs only. The marked asymmetry observed in our case was atypical for WD. No hyper- and hypointensities in the basal ganglia in T1- and T2-weighted MRI images typical for WD were seen in our patient. However, with disease progression, our patient did develop facial dystonia and dysarthria, which are common symptoms in WD.

High heart rate variability observed in our patient might be a sign of autonomic dysfunction observed in WD patients [56]. However, a comprehensive evaluation of autonomic function in WD is lacking.

Cognitive abnormalities in WD occur mainly with regard to attention, executive functions, and memory, while in our patient, despite the presence of mild executive deficits, language impairment and apraxia were predominant. Attention was not tested in detail as speech and limb apraxia made the assessment of processing speed unfeasible.

The *SETX* gene codes for senataxin, a ubiquitously expressed protein that functions as a DNA/RNA helicase, that is crucial for RNA processing, DNA damage response, neurogenesis, and autophagy [57,58]. The *SETX* gene may be considered a cross-disease gene as its mutations cause distinct neurodegenerative phenotypes. Dominant mutations in *SETX* cause juvenile-onset amyotrophic lateral sclerosis type 4 (ALS4), while recessive mutations are associated with ataxia-oculomotor apraxia 2 (AOA2) characterized by cerebellar ataxia, oculomotor apraxia, and axonal sensorimotor neuropathy [57,59]. *SETX* mutations have been detected also in Charcot Marie Tooth (CMT), distal hereditary motor neuropathy (dHMN) [60,61], childhood apraxia of speech [62], and Alzheimer’s disease [63]. None of the symptoms inherent to ALS4 or AOA2 have been observed in our patient.

We observed decreased *SETX* mRNA levels in the patient’s fibroblasts suggesting loss-of-function as the underlying mechanism (Figure 2D). The *SETX* variant c.2385_2387delAAA detected in our patient (Table 2) altered the splicing pattern of exon 10 (Figure 2), generating the aberrant isoform with the 358 bp fragment skipped form exon 10, which was further degraded by the nonsense-mediated decay (Figure 2A, Appendix A). However, as evidenced by sequencing of the main band (Figure 2A, line 6; Appendix A), mRNA carrying the original 3 bp deletion was also generated. It cannot be excluded that the isoform with the original 3 bp deletion also contributes to the decreased levels of *SETX* mRNA possibly affecting the efficiency of downstream processes such as mRNA stability, maturation, transport, or translation. The production of two abnormal transcripts from the mutated allele was described previously [64,65]. For example, neurofibromatosis type 1 (NF1) mutations generated different proportions of mutated transcripts i.e., transcripts containing the exon with the original mutation and those with skipped exon [64,66]. It is worth noting that splicing of exceptionally long exons such as *SETX* exon 10 (4176 bp, Figure 2C, Appendix A), requires precise exon definition to ensure its proper inclusion [67]. Functional variants, such as c.2385_2387delAAA may deregulate this process.

The *SORL1* (sortilin-related receptor 1) gene encodes a transmembrane protein, named sortilin-related receptor (SORLA), involved in endo-lysosomal processes, amyloid precursor protein (APP) sorting and in the degradation of amyloid-beta (Ab) peptide, responsible for Alzheimer’s disease (AD) pathology [68,69]. Initially, both common and rare *SORL1* variants have been associated with AD [70,71,72], while recently, deleterious variants have been also detected in frontotemporal lobar degeneration (behavioral variant and primary progressive aphasia), and dementia with Lewy bodies (DLB) [28,63,73,74,75]. For this reason, *SORL1* is considered a cross-disease gene.

A recent functional study revealed that many *SORL1* rare variants resulted in altered maturation and cellular trafficking of the SORLA protein [76]. The V118M variant detected in our patient localized between two previously characterized mutations S114R and S124R in SORLA protein [76]. The S124R mutation showed a maturation profile similar to the wild-type protein, and the S114R mutation led to the decreased production of a mature, glycosylated form of the protein, which resulted in inefficient transport to the cell membrane [76]. It is thus plausible that also V118M variant may affect maturation defects and cellular localization of SORLA since position 118 in the protein is strongly conserved (phyloP100way).

The *FOXP1* (Forehead Box P1) gene encodes a transcription factor regulating tissue and cell-type-specific gene transcription during development and adulthood [77]. Pathogenic variants in this gene cause neurodevelopmental speech disorders with overlapping apraxic and dysarthric features, predominant motor programming problems, and poor speech intelligibility [78]. Children and adolescents with *FOXP1* syndrome present with considerable delays in language milestones [79]. Expressive deficits may be accompanied by comparable receptive problems in patients with missense variants [78] (expressive language > receptive language). Delays may also encompass motor and intellectual functions [80].

Given the frequency of the *FOXP1* c.1762G>A variant identified in our proband and low scores of various predictive algorithms, ACMG = 0.0188, SIFT = 0.142; PhastCons = 1.0; GERP = 6.17; MCAP = 0.0582, it seems to be benign. However, because our patient had predominant and pronounced speech impairment, characterized by both speech apraxia and—later in the disease course—also spastic dysarthria—it cannot be excluded that *FOXP1* rare variants, such as c.1762G>A, in a particular genetic or epigenetic context, can lead to progressive speech apraxia in adulthood. Such a hypothesis requires further studies. Of note, genetic variations in the *FOXP1* homolog and *FOXP2* gene have been previously found to modulate language performance in FTLD [81].

### 4.2. The significance of Motor Speech Disorders for Delineating the Patient’s Complex Clinical Phenotype

Apraxia of speech (AOS), a motor speech disorder with effortful speech, impaired sequencing of articulatory gestures, and sound-based errors, distinct from both dysarthria and aphasia, is very likely to co-occur with non-fluent aphasia. It may be both neurodevelopmental (childhood-onset) or progressive. In most cases, progressive AOS accompanies PNFA or CBS.

Heterogenous criteria are used in clinical practice to diagnose AOS [82]. In the older literature, apraxia of speech may be misnamed as dysarthria since isolated primary progressive apraxia of speech (PPAOS) was recognized in the context of neurodegenerative diseases, especially CBS, relatively recently [17].

In WD, mixed dysarthria is often observed [83], but dystonic dysarthria associated with orofacial dystonia was also reported [84]. In *ATP7B* heterozygous mutation carriers dysarthria has been reported [85]. In our case, orofacial dystonia appeared late in the disease course. Initially, the patient struggled to perform vertical tongue movements which corresponded to mild oral apraxia. Progressive AOS in our patient was present in the context of full-blown progressive-non fluent aphasia, accompanied by aprosody and later in the disease course, also by spastic dysarthria. Ruggeri et al. reported the presence of dysarthria (not further specified) in 6 out of 11 cases with progressive aphasia and speech apraxia in CBS [86]. Only in 1 out of 12 cases was dysarthria unaccompanied by speech apraxia.

Detailed clinical case descriptions with very long follow-ups and neuropathology data are unfortunately rarely reported. Of note, Tetzloff et al. provided a 10-year-long observation of the PPAOS case with autopsy-confirmed CBD [87]. In this case, spastic dysarthria was added to the AOS in the 8th year of the observation period, about 1–2 years prior to the patient’s death. At the same time, CBS symptoms became prominent.

In the context of *FOXP1* findings, it remains to be established if *FOXP1* abnormalities may influence the onset or pattern of progression of progressive AOS in adults.

### 4.3. Altered Mitochondrial Architecture

Finally, we provide evidence of an altered mitochondrial network in the patient (less branched mitochondria with decreased connectivity) compared to control individuals. As mentioned before, all four genes, *ATP7B, SETX, SORL1*, and *FOXP1*, have been associated with mitochondrial dysfunction [41,42,43,44], a common hallmark of many neurodegenerative conditions including PD and FTLD. Thus, we can hypothesize that the interaction among the four genes may contribute to altered mitochondria architecture in the patient.

### 4.4. Limitations of the Study

The patient’s clinical diagnosis was not supported by the analysis of WD metabolic markers, cerebrospinal fluid (CSF) biomarkers, dopamine transporter SPECT [88], or amyloid/tau PET imaging, which could have helped to further elucidate the etiology [89]. Moreover, the autopsy was not performed.

As DNA from the patient’s family members was not available, the co-segregation of the identified variants with the disease phenotype in the family has not been established.

## 5. Conclusions

The large-scale sequencing techniques have the potential to facilitate a diagnosis for intractable dementia cases with heterogenous clinical manifestations. Our results are in line with those of Ciani et al., who found that 50% of sporadic patients from the FTLD spectrum showed at least one rare missense variant in AD, PD, ALS, and LBD-associated genes [28]. Our results spoke in favor of the oligogenic inheritance in complex neurodegenerative phenotypes, such as CBS-NAV.

To our knowledge, this is the first reported case of the CBS-NAV phenotype with a single heterozygous *ATP7B* mutation. While parkinsonian phenotypes have been previously linked to *ATP7B* heterozygotes, we also described other variants of potential functional impacts in well-known disease-related genes, such as *SETX, SORL1*, and *FOXP1*. Of note, these three latter genes have been associated with speech and/or language deficits. The dissection of the molecular mechanisms underlying possible complex interactions among these genes would be a great challenge for the future.

## Figures and Tables

**Figure 1 genes-13-02361-f001:**
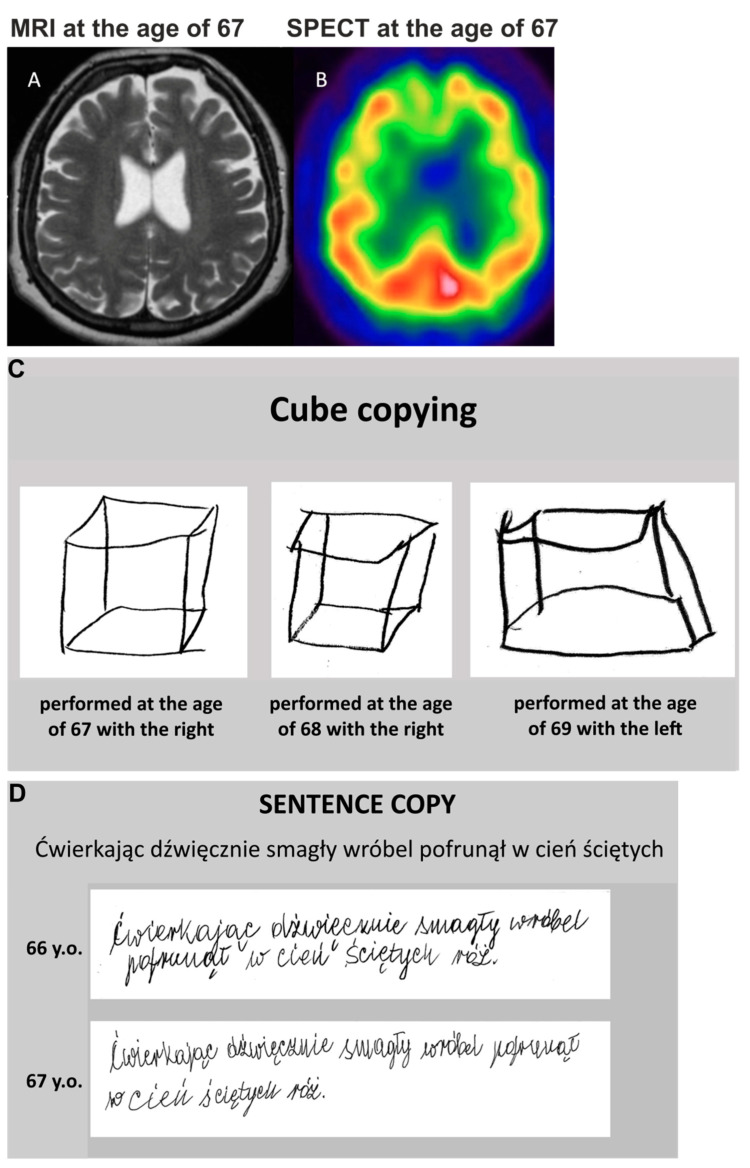
Neuroimaging and neuropsychology of the patient. (**A**). MRI at the age of 67 y.o.—bilateral, moderate frontal lobe atrophy; (**B**) SPECT- bilateral frontal and left parietal hypoperfusion; (**C**) cube copying task performed at 3 neuropsychological assessments. Even at the last assessment, when the patient drew with her left hand, the attempt to depict three-dimensionality is evident; (**D**) preserved sentence copying. Despite marked right upper limb rigidity, the patient was initially able to write.

**Figure 2 genes-13-02361-f002:**
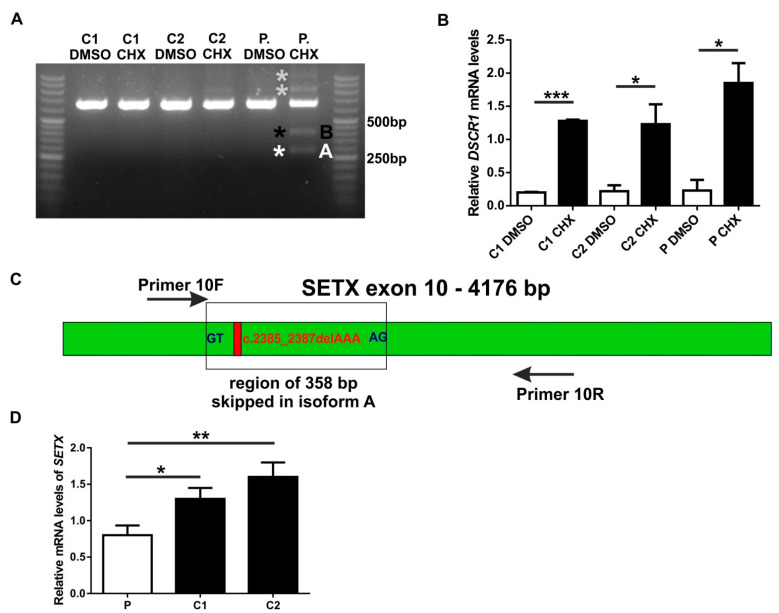
(**A**). Visualization of cDNA analysis on agarose gel. Fibroblasts from the patient (P) and controls (C1, C2) were treated with DMSO (lines 1, 3, 5) or cycloheximide (CHX) (line 2, 4, 6) for 10 h to block nonsense-mediated decay. White asterisk indicates aberrant isoform (*A) with partial skipping (358 bp) of *SETX* exon 10 in the patient’s fibroblasts, arising due to the presence of variant c.2385_2387delAAA, as evidenced by sequencing of band A (see Appendix A). Sequencing of the excised band *B (Figure 2A, lane 6) revealed that it is an artifact; (**B**) mRNA transcribed from *DSCR1* gene (the Down syndrome critical region 1), a known NMD target, was used as a positive control to confirm NMD inhibition; (**C**) schematic representation of *SETX* exon 10, black empty square denotes 358 bp sequence skipped from exon 10 carrying c.2385_2387delAAA variant. Black arrows correspond to primers used for isoform analysis; (**D**) real-time PCR demonstrated lower levels of *SETX* mRNA in the patient compared to healthy controls (Appendix A). P was calculated in two-tailed *t*-test, * *p* < 0.05, ** *p* < 0.01, *** *p* < 0.001.

**Figure 3 genes-13-02361-f003:**
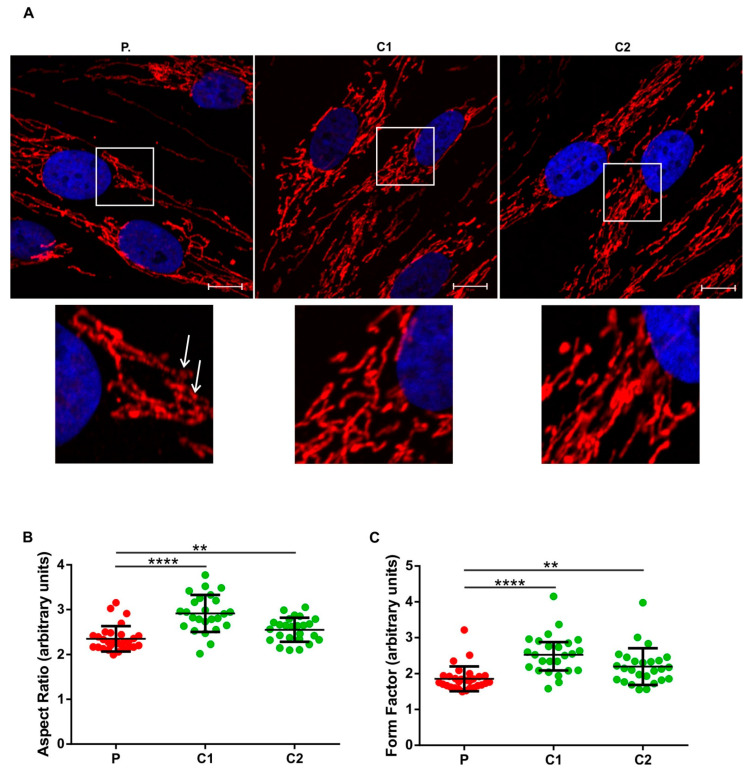
(**A**) The mitochondrial network in the patient’s fibroblasts (P) compared to neurologically healthy, age-matched controls (C1, C2). Area delineated with the white square in the upper panel is enlarged in the lower panel. White arrows point to round, fragmented mitochondria. Red: Hsp60—mitochondria, blue: DAPI—nucleus; representative images are shown; (**B**) decreased form factor and (**C**) decreased Aspect Ratio in the patient’s mitochondria compared to controls. Aspect Ratio < 0.0001, Kruskal–Wallis test; Form Factor < 0.0001, Kruskal–Wallis test; P for differences between individual cell lines: P vs. C1, P vs. C2 were calculated with unpaired t test with Welch’s correction, **** < 0.0001, ** < 0.01. All images were taken at 63× magnification. Each dot corresponds to a measurement from one cell. Upper panel scale bar = 10 µm.

**Table 1 genes-13-02361-t001:** Statistics of mapped reads (DNA).

Number of Mapped Reads (in Millions)	Proportion of Sequenced Reads (in %)	PCR Duplicates (in %) *	Median Insert Size (in bp)	Average Coverage
198.435	86.7	12.44	206	213.5

(*) PCR duplicates are duplicated reads that are removed during mapping. In general, a high PCR duplicate rate correlates with low sample quality.

**Table 2 genes-13-02361-t002:** The rare variants of.the highest impact on the disease phenotype according to bioinformatic priotitization.

Gene	HGVS ^1^ DNA/Exon/Protein/rs	Predicted Effect	MAF gnomAD	CADD * Phred	ACMG Criteria
*ATP7B*	c.3207C>A exon 14 p.His1069Gln rs76151636 heterozygous	missense	0.001019	24	PS1, BP4, BP5, PP3 likely pathogenic
*SETX*	c.2385_2387delAAA exon 10 p.Ile795_Lys796delinsMet rs755971927 heterozygous	in frame deletion	0.0000329	NA	PM2, PM4, PP4 uncertain significance
*SORL1*	c.352G>A p.Val118Met exon 2 rs749389644 heterozygous	missense	0.00001060	28.1	PM2, PP3 uncertain significance
*FOXP1*	c.1762G>A p.Ala588Thr exon 20 rs202173892 heterozygous	missense	0.000318	21.7	BP5 pathogenic/likely pathogenic or of uncertain impact

^1^ Variant nomenclature according to Human Genome Variation Society. *—Combined Annotation-Dependent Depletion score Phred-scaled [38].

## Data Availability

The data presented in this study are sequencing data of human samples thus cannot be openly shared due to personal data protection.

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
