# Peer review of "A Patient with Corticobasal Syndrome and Progressive Non-Fluent Aphasia (CBS-PNFA), with Variants in ATP7B, SETX, SORL1, and FOXP1 Genes"

_genes, 2022, doi:10.3390/genes13122361_

Round 1

Reviewer 1 Report

In this paper, the authors have analyzed the phenotype-genotype correlations in a patient diagnosed with early onset CBS-NAV with predominant apraxia of speech, performing whole exome sequencing and identifying rare single heterozygous variants in ATP7B, SETX, SORL1, and FOXP1 genes. The paper is well written and compelling and provides exhaustive clinical and genetic information.

I have only some minor suggestions for the authors:

-        In the Table 2, it could be helpful to include which exon is affected by the mutation

-        In the Figure 2A, please insert a description for the * B band in the legend

-        The legend of the Figure 2B is not very clear

-        In the Figure 3A, the scale bar is difficult to read, please improve its quality

-        Regarding the in-frame deletion in SETX gene, the authors demonstrated the presence of an aberrant splicing variant and the decrease of wild-type isoform, suggesting a loss-of-function mechanism. However, sequencing results of the main band excised from agarose gel confirmed the presence of the heterozygous SETX deletion, as shown in Figure S5. Thus, the presence of the deletion seems to lead to both an aberrant splicing and the mRNA showing the in-frame deletion. The authors should comment this point.

Author Response

We thank the Reviewer for the insightful comments.

We have addressed all the issues raised by the Reviewers and corrected the manuscript according to their suggestions. We believe that these changes improved the quality and clarity of the manuscript.

Please find enclosed the revised version of our manuscript entitled “A patient with the corticobasal syndrome, progressive non-fluent aphasia (CBS-PNFA) and variants in ATP7B, SETX, SORL1, and FOXP1 genes” by Katarzyna Gaweda-Walerych, Emilia Jadwiga Sitek, MaÅ‚gorzata Borczyk,  Ewa NarożaÅ„ska,  Bogna Brockhuis, MichaÅ‚ KorostyÅ„ski, MichaÅ‚ Schinwelski, Mariusz SiemiÅ„ski, JarosÅ‚aw SÅ‚awek, and Cezary Zekanowski.

The changes in the manuscript have been highlighted in yellow.

Reviewer  1

In this paper, the authors have analyzed the phenotype-genotype correlations in a patient diagnosed with early onset CBS-NAV with predominant apraxia of speech, performing whole exome sequencing and identifying rare single heterozygous variants in ATP7B, SETX, SORL1, and FOXP1 genes. The paper is well written and compelling and provides exhaustive clinical and genetic information.

I have only some minor suggestions for the authors:

Reviewer  1 -        In the Table 2, it could be helpful to include which exon is affected by the mutation

We have introduced this information in column 2 of Table 2 according to the Reviewer’s suggestion.

Reviewer  1 -        In the Figure 2A, please insert a description for the * B band in the legend

We have introduced the information in the legend of Fig 2A, according to the Reviewer’s suggestion.

Reviewer  1 -        The legend of the Figure 2B is not very clear

mRNA transcribed from DSCR1 gene (Down syndrome critical region 1) is a known nonsense-mediated decay (NMD) target (Gong 2009, Wittmann 2006). For this reason, it is commonly used as a positive control in NMD inhibition experiments, since DSCR1 mRNA levels increase upon NMD inhibition with cycloheximide (Gaweda-Walerych 2016, 2018). We added the explanation in the text (Results section), in the Fig 2B legend, and provided appropriate citations.

Reviewer  1 -        In the Figure 3A, the scale bar is difficult to read, please improve its quality

We thank the Reviewer for pointing this out, we corrected the scale bar in the Fig 3A.

Reviewer  1 -        Regarding the in-frame deletion in SETX gene, the authors demonstrated the presence of an aberrant splicing variant and the decrease of wild-type isoform, suggesting a loss-of-function mechanism. However, sequencing results of the main band excised from agarose gel confirmed the presence of the heterozygous SETX deletion, as shown in Figure S5. Thus, the presence of the deletion seems to lead to both an aberrant splicing and the mRNA showing the in-frame deletion. The authors should comment on this point.

We thank the Reviewer for this observation. The alternative splicing process has variable efficiency and variable outcomes. Thus, it can be expected that only the proportion of the SETX pre-mRNA containing the 3bp deletion undergoes the aberrant splicing that skips the 358 bp fragment. Skipping of 358 bp from exon 10 during splicing causes a frameshift and introduces premature termination codon (PTC), such isoform is further degraded by the NMD mechanism.

In parallel, the mRNA carrying the original 3bp deletion is also generated from the mutated allele. It can possibly exert harmful effects on the cell through other molecular mechanisms. For example, it can affect downstream processes such as mRNA stability, maturation, transport, or translation. Also, it cannot be excluded that the isoform with 3bp deletion contributes to the decreased levels of SETX mRNA.

We added additional comments regarding this issue in the abstract, Result section, and in Discussion section, on page 11. They are highlighted in yellow.

Reviewer 2 Report

Authors provided an original article suggesting new insights on clinical spectrum and phenotypic-genetic correlations of neurodegenerative disorders.

It was a pleasure to revise this article: results are clearly presented and supported by functional studies; clinical assessments of the patient is very well detailed.

However revisions are suggested in order to make the article more complete and accurate. In particular:

Ø  The introduction needs to be expanded, being lacking in some points:

·        - Line 45-49: this part should be extended with a more detailed description of known genetic causes of CBD and PNFA

·         -Line 54: In the cited article (Armstrong MJ et al., 2013) the “variant with non-fluent/agrammatic progressive aphasia” was not mentioned as CBD-NAV. The “nonfluent/agrammatic variant of primary progressive aphasia” was reported as naPPA. Could you please check and possibly cite the reference in which the CBD-NAV is mentioned? Similarly Line 161-162

·         -Line 57-59: the aim of the study should be more detailed, for example citing more references supporting the poligenic etiology of overlapping syndromes with early onset; moreover the role of WES in this context (with its advantages and potential) should be discussed

Ø  Regarding results:

·         -Line 168-170: Could you please also provide any further information about family history? In the Supplementary Materials, a figure with family pedigree could also be appropriate

·         -Line 224: Probably the authors are referring to Table 2

·        - Line 225: Please specify with which type of analysis/method the C9orf72 expansion has been excluded. Furthermore, the reference cited (17) does not appear very appropriate

Ø  In the discussion:

·         -Line 307: Please specify which elements indicate that in the patient there is a “subclinical dysregulation of copper metabolism”

·         -In the discussion related to ATP7A gene, the data “high heart rate variability” (reported in the results paragraph) should be commented (in consideration to known autonomic dysfunctions of WD patients)

·         -Line 402-403: The sentence “we propose that the interaction among the four genes may contribute to altered mitochondria architecture in the patient” should be supported by evidence or, alternatively, should be better argued (by including molecular mechanisms underlying possible complex interactions)

-Line 409-410: As suggested for the results paragraph, more family history information should be provided. Is no family member available for a segregation analysis of the identified variants?

Ø  In general in the introduction and in the discussion, I suggest to expand following points:

·         -poligenic background of neurodegenerative diseases underlying clinical variability

·         -the relevance of WES studies in this context

Author Response

We thank the Reviewer for the insightful comments.

We have addressed all the issues raised by the Reviewers and corrected the manuscript according to their suggestions. We believe that these changes improved the quality and clarity of the manuscript.

Please find enclosed the revised version of our manuscript entitled “A patient with corticobasal syndrome, progressive non-fluent aphasia (CBS-PNFA) and variants in ATP7B, SETX, SORL1, and FOXP1 genes” by Katarzyna Gaweda-Walerych, Emilia Jadwiga Sitek, MaÅ‚gorzata Borczyk,  Ewa NarożaÅ„ska,  Bogna Brockhuis, MichaÅ‚ KorostyÅ„ski, MichaÅ‚ Schinwelski, Mariusz SiemiÅ„ski, JarosÅ‚aw SÅ‚awek, and Cezary Zekanowski.

The changes in the manuscript have been highlighted in yellow.

 Reviewer  2

Authors provided an original article suggesting new insights on clinical spectrum and phenotypic-genetic correlations of neurodegenerative disorders.

It was a pleasure to revise this article: results are clearly presented and supported by functional studies; clinical assessments of the patient is very well detailed.

However revisions are suggested in order to make the article more complete and accurate. In particular:

Reviewer  2 Ø  The introduction needs to be expanded, being lacking in some points:

 Reviewer  2- Line 45-49: this part should be extended with a more detailed description of known genetic causes of CBD and PNFA

We thank the Reviewer for this comment. We have extended the Introduction section, according to the Reviewer’s suggestions. The added text is highlighted in yellow.

Reviewer  2 -Line 54: In the cited article (Armstrong MJ et al., 2013) the “variant with non-fluent/agrammatic progressive aphasia” was not mentioned as CBD-NAV. The “nonfluent/agrammatic variant of primary progressive aphasia” was reported as naPPA. Could you please check and possibly cite the reference in which the CBD-NAV is mentioned? Similarly Line 161-162

NAV phenotype is indeed mentioned in the cited paper in Table 5 (Diagnostic criteria for corticobasal degeneration p. 501). The authors of the criteria used in their paper both terms: naPPA and NAV. We decided to use only one of these two terms so as to avoid confusion. As Armstrong et al. have not used the term PNFA and we wanted to mention in our report at least one of the terms that are present in the current diagnostic criteria of CBD.

Reviewer  2-Line 57-59: the aim of the study should be more detailed, for example citing more references supporting the poligenic etiology of overlapping syndromes with early onset; moreover the role of WES in this context (with its advantages and potential) should be discussed

According to the Reviewer’s suggestion, we have provided references regarding the raised issues in the introduction section. The added text and references are highlighted in yellow.

Reviewer  2 Ø  Regarding results:

Reviewer  2 -Line 168-170: Could you please also provide any further information about family history? In the Supplementary Materials, a figure with family pedigree could also be appropriate

We provided information on the patient’s father who showed similar symptoms (page 4). The patient’s father has been adopted. For this reason, no patient’s grandparents were available on this side of the family. We have omitted this sensitive information because it could help identify the patient. Unfortunately, more information on the pedigree was not available.

Reviewer  2 -Line 224: Probably the authors are referring to Table 2

We thank the Reviewer for pointing out this typing error. We corrected the numbering of the table.

Reviewer  2 - Line 225: Please specify with which type of analysis/method the C9orf72 expansion has been excluded. Furthermore, the reference cited (17) does not appear very appropriate

According to the Reviewer’s suggestion, we have added an appropriate description along with the original article citation in materials and methods (highlighted in yellow, page 3).

Reviewer  2 Ø  In the discussion:

Reviewer  2-Line 307: Please specify which elements indicate that in the patient there is a “subclinical dysregulation of copper metabolism”

We have provided comprehensive data on the effects of single heterozygous mutations in the ATP7B gene in Table S10 summarizing the available literature. The evidence gathered there suggests that single heterozygous mutations in the ATP7B gene were in some cases associated with “subclinical dysregulation of copper metabolism”. However, in our case, the analysis of biochemical markers of WD (e.g. decreased ceruloplasmin or elevated copper levels) has not been performed. This is described as a limitation of the study in section 4.4.

For this reason, we can only speculate that there might have been a  “subclinical dysregulation of copper metabolism” due to the presence of p.H1069Q mutation in the patient.

We have changed the expression “it cannot be excluded” to “it can be speculated” in the following sentence on page 10:

“Given the mounting evidence of the possible involvement of single heterozygous ATP7B mutations in phenotypes with parkinsonism (Table S10), it can be speculated that p.H1069Q mutation in our CBS-NAV patient has led to the subclinical dysregulation of copper metabolism.”

Reviewer  2 -In the discussion related to ATP7A gene, the data “high heart rate variability” (reported in the results paragraph) should be commented (in consideration to known autonomic dysfunctions of WD patients)

We thank the Reviewer for pointing out this aspect. As reported by Li et al. (doi: 10.3389/fphys.2017.00778.) patients with Wilson's disease showed higher heart rates than control subjects.  However, a comprehensive evaluation of autonomic function is lacking. We have commented on this observation in the discussion section (page 10, highlighted in yellow):

“High heart rate variability observed in our patient might be the sign of autonomic dysfunction observed in WD patients [49]. However, comprehensive evaluation of autonomic function in WD is lacking.

Reviewer  2-Line 402-403: The sentence “we propose that the interaction among the four genes may contribute to altered mitochondria architecture in the patient” should be supported by evidence or, alternatively, should be better argued (by including molecular mechanisms underlying possible complex interactions)

We have also provided appropriate references on the association between all four genes: ATP7B, SETX, SORL1, and FOXP1 and mitochondrial function.

We are convinced that stating hypotheses based on the obtained results is important for defining research questions and delineating future research objectives.

We have changed the sentence to show it refers to a research hypothesis and not a conclusion from the obtained data.

Reviewer  2-Line 409-410: As suggested for the results paragraph, more family history information should be provided. Is no family member available for a segregation analysis of the identified variants?

Unfortunately, no family member was available for a segregation analysis which is highlighted in section 4.4 Limitations of the study.

We provided information on the patient’s father who showed similar symptoms (page 4). The patient’s father has been adopted. For this reason, no patient’s grandparents were available on this side of the family. We have omitted this sensitive information because it could help identify the patient.

Reviewer  2 Ø  In general in the introduction and in the discussion, I suggest to expand following points:

  •         -poligenic background of neurodegenerative diseases underlying clinical variability
  •         -the relevance of WES studies in this context

We thank the Reviewer for raising these issues. We have added comments in the Introduction section regarding the raised issues. The changes are highlighted in yellow.
